# Bending Fatigue Behaviour and Fatigue Endurance Limit Prediction of 20Cr2Ni4A Gear Steel after the Ultrasonic Surface Rolling Process

**DOI:** 10.3390/ma14102516

**Published:** 2021-05-12

**Authors:** Zhiyuan Wang, Yangfei Huang, Zhiguo Xing, Haidou Wang, Debin Shan, Fengkuan Xie, Jiming Li

**Affiliations:** 1School of Materials Science and Engineering, Harbin Institute of Technology, Harbin 150001, China; reincarnational@163.com (Z.W.); shandebin@hit.edu.cn (D.S.); 2National Key Lab for Remanufacturing, Army Academy of Armored Forces, Beijing 100072, China; huangyanfei123@126.com (Y.H.); xiefengkuan@126.com (F.X.); lijm0215@163.com (J.L.); 3National Engineering Research Center for Remanufacturing, Army Academy of Armored Forces, Beijing 100072, China

**Keywords:** ultrasonic surface rolling treatment, 20Cr2Ni4A carburizing gear steel, three-point bending fatigue test, microscopic mechanism

## Abstract

To study the effect of the surface properties on the bending fatigue performance of heavy-duty gear steel, the authors of this paper used the ultrasonic surface rolling process (USRP) to strengthen 20Cr2Ni4A carburized gear steel. USRP is a novel technique in which the ultrasonic technology is incorporated into the concept of conventional deep rolling. In this study, we illustrated how the surface properties and cross-section mechanical property influence the three-point bending fatigue life of the samples before and after USRP treatment. At the same time, the predicted failure probability-stress-number of cycles (*P*-*S*-*N*) curve was drawn, and the fatigue fracture was analysed. The results show that the fatigue limit increased from 651.36 MPa to 918.88 MPa after USRP treatment. The fatigue source is mainly from the sample interior or surface scratches, and the fatigue performance is positively correlated with the results of the material surface roughness, surface residual stress and surface hardness. At the same time, combined with the change in the phase structure, dislocation structure, residual stress and hardness of the cross section of the material, it is found that the USRP process turns the steel into a gradient material with five layers. Finally, the coupling mechanism between the ultrasonic surface strengthening deformation layer and the carburized layer of 20Cr2Ni4A carburized gear steel is presented, and the grain structure distribution diagram of the section of the 20Cr2Ni4A model after surface strengthening treatment was simulated. The mechanism that influenced the fatigue performance after USRP treatment is explained from the perspectives of the surface and cross section of the samples.

## 1. Introduction

The gear transmission system is a key component in the fields of aerospace, energy, transportation, and large machinery transmission, and its reliability affects the safety of the entire transmission system and even of the entire mechanical system [1]. After years of research, there are still three outstanding problems of heavy-duty gear manufacturing. They are short life, heavy structural load and poor reliability [2]. Therefore, the core problem of manufacturing high-end heavy-duty gears is how to improve their fatigue performance while ensuring reliability and light weight [3]. This requires a heavy-duty gear with a high-strength surface and a high-toughness core. The key to this requirement is whether the surface structure of the gear teeth can effectively improve the gear bearing capacity and bending fatigue life. For the strengthening of the gear tooth surface, lots of surface treatments, such as plasma spraying [4], acid etching [5], sandblasting [6], high energy shot peening [7], surface mechanical attrition treatment [8], ultrasonic shot peening [9,10], ultrasonic surface rolling (USR) [11], surface ultrasonic impact [9], etc., were applied for surface modification or strengthening to better suit some specific situations.

Among the many surface treatment processes, ultrasonic processing has been widely used in the field of gear manufacturing due to its wide processing range, safety, reliability, stability and high efficiency [12]. The ultrasonic surface rolling process (USRP) is the most prominent method to treat the surface properties of aircraft landing gear steel; it combines static rolling and ultrasonic impact on the surface of the material to improve its overall performance. Based on the deformation strengthening theory [13], after ultrasonic surface rolling treatment, the grains on the metal surface undergo elastoplastic deformation, and the metal organization becomes denser, which improves the strength of the workpiece. Dan Liu et al. [14] used USRP to treat martensitic precipitation hardened (17-4PH) stainless steel and obtained a gradient nanocrystalline structure with a thickness of 650 μm. The grain size showed regular large-small-large changes, while the high-low-high dislocation density changed with increasing depth. Tolga Bozdana [15] used USRP for Ti-6Al-4V samples and found that processing thin parts did not cause severe deformation of the parts because the pressure applied on the surface of the parts was small. At the same time, residual compressive stress of approximately −900 MPa was obtained at a depth of 0.2 mm from the surface. QL Zhang et al. [16] used USRP to improve the surface properties of 17-4PH stainless steel and carried out a detailed study of the microstructure and mechanical properties of 17-4PH stainless steel after USRP treatment. The nanograin layer significantly improved the surface properties (surface roughness, microhardness, and residual compressive stress), and the abrasion resistance and corrosion resistance of the samples after USRP treatment significantly improved. Sik Pyun et al. [17] compared the stress and number of cycles (*S*-*N*) curves from a rotational bending test to determine the optimal process parameters with the greatest fatigue strength. The surface roughness, hardness and residual compressive stress of the three samples were analysed, and the fracture surface was analysed by scanning electron microscopy (SEM). It was shown that an ultrasonic nano surface modification (UNSM) treatment prevented the premature failure of hot-rolled stainless steel shear pins, prolonged the service life and reduced replacements. Based on this, it was found that the USRP method has the following advantages:(1)It provides a reliable process and smooth surface. Once semi-finished samples are processed, the surface roughness can be greatly reduced and the residual stress can be introduced into the workpiece.(2)It provides a low force on the workpiece. The acting force is elastic and has no adverse effect on the machine tool.(3)As there is no high-temperature process, the gear material hardly undergoes structural transformation, and the original fine structure formed by heat treatment or forging can be maintained.(4)It provides uniform strengthening and stress control through the process parameters (static pressure, amplitude, step and processing speed, for example) and can control the strengthening layer depth. Moreover, there is a continuous transition between the reinforced layer and the matrix without stripping.

Although domestic and foreign scholars have achieved results on the research of USRP, the following problems still exist: (1) the mechanical dynamic response mechanism of gear steel under USRP treatment has not yet been clarified; (2) the coupling mechanism between the deformation strengthening layer and carburized layer remains to be studied; and (3) from the perspective of the relationship between the microscale mechanism and the macroscale performance, clarification of the influence of USRP on the bending fatigue properties of carburized gear steel has rarely been reported. Based on these three points, the mechanism that is operative during the USRP treatment of carburized gear steel is studied in this paper.

## 2. Materials and Methods

### 2.1. Materials

In this paper, 20Cr2Ni4A steel, which is commonly used in heavy-duty transmission gears, was used as the research object. A TXC01 direct-reading spectrometer was used to determine the composition of the steel, and the range of components is presented in Table 1.

After the drawing forming of 20Cr2Ni4A steel, the blank was finished and processed into bone type standard tensile sample for tensile test. Standard tensile samples were carried out on INSTRON5985 electronic universal material testing machine (INSTRON (Shanghai) Test Equipment Trading Co., Ltd., Shanghai, China) according to GB/T 228-2002 [18], with a calibration distance of 20 mm and a strain rate of 10^−2^/s, to determine the tensile properties of steel. The tensile strength *σ*_b_ of the 20Cr2Ni4A material is 1043 MPa, the yield strength *σ*_0.__2_ is 732 MPa, the elongation δ is 20%, the reduction in area ψ is 65%.

To simulate the actual use of heavy gears, we used the heat treatment process that is the same as that used for heavy-duty gears to heat treat steel. The process involves normalizing, carburizing, high-temperature tempering, quenching, and low-temperature tempering, and the specific temperatures are presented in Table 2.

Finally, the size of three point bending fatigue samples was as follows: height *h* = 15 mm, thickness *b* = 30 mm, sample length *S* = 210 mm, span *L* = 120 mm, diameter of roller R = 30 mm, and the rounding radius set during processing was 3 mm. The detailed shape and dimensions are shown in Figure 1.

### 2.2. Experimental Procedures

#### 2.2.1. Ultrasonic Surface Rolling Process

The USRP treatment was performed on a Hawken computer numerical control (CNC) milling machine. The cutter head was a carbide ball (the schematic diagram is shown in Figure 2) with a diameter of 8 mm. Before the USRP treatment, the carburized sample was milled on a numerically controlled machine tool to remove the material surface oxide caused by the carburizing heat treatment. Large surfaces were selected as the processing surface; the length of the USRP threated surface was 100 mm, and the width was 30 mm. The sample is fixed on the fixture, and the feed motion is on the tool. To ensure that the direction of the shear force in the USRP area was consistent during processing, a unidirectional repetitive processing path was used, as shown in Figure 2. To reduce the influence of the adiabatic temperature rise during processing, a mixture of kerosene and mechanical oil was used as the lubricating fluid. They were mixed in a ratio of 1:3, which removed the heat and debris generated during processing.

During the USRP treatment, parameters such as the static load, amplitude, line speed, step length, and the number of processing times will affect the strengthening effect. In general, the greater the static load and amplitude in the USRP process, the more obvious the strengthening effect is, and the greater the surface residual compressive stress will be. However, too much static load and amplitude can increase surface roughness. An excessively large static load and amplitude will increase surface roughness. With the increase in linear velocity, the processing efficiency will be improved, but too fast linear velocity will also increase the surface roughness. Taking the above factors into consideration, the process parameters were selected as presented in Table 3. Static load was provided by an air compression device (with a bore diameter of 50 mm).

#### 2.2.2. Surface Roughness Test

The surface and cross-sectional microstructures of the four samples before and after USRP treatment were observed with confocal laser scanning microscopy (LSCM) on a LEXTOLS4000 instrument (Olympus Corporation, Shibuya-ku, Tokyo, Japan) and SEM on a ZEISS Supra 55 field emission scanning electron microscope (Carl Zeiss AG, Jena, Germany). The three-dimensional morphologies of the 4 samples before and after the USRP treatment are shown in Figure 3. The 0^#^ sample shows the morphology of the sample without USRP treatment. The 1^#^, 2^#^ and 3^#^ samples show the morphologies after the USRP treatment. In comparison, it can be seen that the surface texture of the processed material tends to be uniform due to the friction reduction in the USRP. At the same time, the surface tool marks after milling are almost pressed due to the plastic flow of the metal. Moreover, as the static load level increases, the surface texture tends to be smoother, but it can be seen in Figure 3 that when the static load reaches 1963 N, the surface scratches become deep because the plastic deformation of the surface is too large due to the excessive static load, resulting in obvious machining cracks.

#### 2.2.3. Surface Microhardness Test

The surfaces and section microhardness of the four types of samples were tested by a Shimadzu HMV-2000 Vickers hardness tester (Shimadzu Enterprise Management (China) Co., Ltd., Hong Kong, China) (the indenter of the durometer is a diamond cone indenter), respectively. After many orthogonal tests, the optimal test parameters were as follows: the pressure applied to the sample was 300 g, pressure holding time was 10 s, and point spacing was 400 μm. Due to the difference in the structure of the sample at different positions, 25 points were randomly selected at different positions in the USRP treatment section to ensure a thorough microhardness measurement, and the average value is reported as the final microhardness of the samples.

#### 2.2.4. Surface Residual Stress Test

An X-ray residual stress measuring device (TEC4000, Stresstech Oy, Helsinki, Finland) was used to measure the size and distribution of the residual stress along the surface and depth of the 20Cr2Ni4A sample. The parameters of the X-ray stress analyser are set as follows: the target material is Cr target, the tube current is 6.7 mA and the tube voltage is 30 kV. ψ angle is −45°–+45°, spot is 3 mm, exposure is 5 s. When measuring along the depth direction, the surface material was stripped layer by layer by electrolytic polishing to avoid the impact of external forces [19]. The device can detect the residual compressive and tensile stress at a depth of 20 μm. During the test, 25 points were evenly distributed on the surface of the sample in the form of a matrix, and each point was tested three times with an interval of 5 min.

#### 2.2.5. Crystal Structure Test

According to the data analysis in Section 2.1, it can be seen that among the four styles, the 1^#^ style was used as a control group, and the performance is compared for the 0^#^ style (without USRP treatment) and the 2^#^ style (standard USRP treatment), and 3^#^ belongs to the USRP over processed group. From a macroscopic perspective, the style of the roughness, surface hardness and residual stress on the fatigue is severe, but from a microscopic perspective, is the same conclusion reached? Based on this question, this paper selects the standard 0^#^ sample (without USRP treatment) and 2^#^ sample (standard USRP treatment) to define the fatigue damage mechanism from a microscopic perspective. Transmission electron microscopy (TEM) was conducted on a Tecnai F20 instrument (American FEI Company, Hillsboro, OR, USA) and used to photograph the microscopic appearance of the surface layer treated by USRP. The XRD samples at the fracture were obtained by an MXP21VAHF X-ray diffractometer. The grain size and distribution of the sample were probed by an FEI quanta 650FEG thermal field emission scanning electron microscope and electron backscatter diffraction (EBSD).

#### 2.2.6. Three-Point Bending Fatigue Behaviour Test

The bending fatigue test used a PLG-300C high-frequency fatigue tester, which is shown in Figure 4a. The maximum average load of the device is ±300 kN, the maximum alternating load is ±150 kN, and the frequency ranges from 70–250 Hz. The fatigue behaviour of the 20Cr2Ni4A steel samples designed in this paper was tested by three-point bending loading [20,21]. The representative 0^#^, 2^#^, and 3^#^ samples were selected for fatigue testing, and the corresponding *S*-*N* and *P*-*S*-*N* curves were obtained. Due to the large discreteness of the fatigue test, 30 samples were taken for a three-point bending fatigue test in each group to ensure the reliability of the data.

The fatigue test was studied by the group comparison test, and the fixture is shown in Figure 4b. At least three stress levels were selected, and 5 valid test data points were obtained at each stress level. Therefore, the 0^#^, 2^#^ and 3^#^ test groups prepared 18 samples for the experiment, respectively (2^#^ samples are shown in Figure 4c). The cyclic characteristic coefficient is defined by the load ratio *r* = *F_min_*/*F_max_* = 0.5. The sinusoidal loading method was adopted, and the frequency is 79 Hz. Since the stress in the middle of the sample (S/2) is the greatest during the fatigue test, the surface treated by USRP is placed downward and within the span range to ensure the accurate measurement of the strengthening effect of USRP treatment.

The average load and alternating load were adjusted to achieve different stress levels to realize pulsating loading. According to the test machine fixture position and chuck size, the bending stress loaded on the pattern was tested. The detailed data are presented in Table 4.

The fracture surface after the test was cleaned with alcohol and observed with a ZEISS Supra 55 field emission scanning electron microscope. Finally, the surface of the sample before the test was analysed to determine the relationship between the integrity and the flexural fatigue performance of the 20Cr2Ni4A steel.

## 3. Experimental Results

### 3.1. Results of the Three-Point Bending Fatigue Performance

Table 4 shows the three-point bending fatigue test data of the samples before and after the USRP treatment. Due to many factors that affect the fatigue test results, such as equipment errors, material nonuniformities, processing deviations, and the environment, the fatigue test results are largely discrete, and the fatigue stress and fatigue life results do not have a strict one-to-one correspondence but are closely related to the survival rate P. The conventional *S*-*N* curve represents the median fatigue life curve that corresponds to a 50% survival rate. The method researched in this paper can cause large deviations in results due to errors in performance improvement. Therefore, the *P*-*S*-*N* curve is used to test the fatigue life and fatigue limit of the three sets of samples, which is the most consistent with the design scheme of this article. At the same time, the *P*-*S*-*N* curve can comprehensively express the relationship between fatigue stress and fatigue life at various reliability levels and determine the degree of life dispersion of the three materials under different stresses. In the calculation of the *P*-*S*-*N* curve, it is generally considered that when life is constant, the material fatigue limit obeys a normal distribution and log-normal distribution. When the stress is constant, the fatigue life obeys a logarithmic distribution and Weibull distribution under low cycles (*N* < 10^6^); when the life reaches high cycle fatigue (*N* > 10^6^), it obeys the Weibull distribution [22]. Therefore, assuming that the data at each group of stress levels conform to the two-parameter Weibull distribution, the distribution function equation is:
(1)$$P\left(N\right)\;=\;1\;-\;\exp\left[-{\left(N/N_{a}\right)}^{\beta }\right]$$
where *P* (*N*) is the failure probability of the fatigue test, N is the fatigue life, *β* is the shape parameter, and *N*_a_ is the scale parameter, which is also the characteristic life of the material. The key to drawing the Weibull distribution curve is to use appropriate methods to estimate the values of the two *N*_a_ and *β* parameters. The most widely used maximum likelihood estimation method is used to estimate the parameters of the Weibull curve. The two-parameter *N*_a_ and *β* estimation equations are shown in Equations (2) and (3):
(2)$$\frac{\sum_{i=1}^{n}N_{i}^{\beta }\ln N_{i}}{\sum_{i=1}^{n}N_{i}^{\beta }}-\frac{1}{n}\sum_{i=1}^{n}\ln N_{i}-\frac{1}{\beta }=0$$
(3)$$N_{a}={\left(\frac{1}{n}\sum_{i=1}^{n}N_{i}^{\beta }\right)}^{\frac{1}{\beta }}$$

Based on Equations (2) and (3) [23], the bending life data of the three samples tested in Section 2.2 under four different bending stresses are calculated(the bending life test results are in the Appendix A), and the estimated values of the two parameters *N*_a_ and *β* presented in Table 5 are obtained.

The predicted failure probability and the number of cycles (*P*-*N*) curve of the material drawn according to Table 5 is shown in Figure 5. From the data in the figure and the table, it can be seen that the *β* value of the 0^#^ sample is smaller than that of both 2^#^ and 3^#^. Among them, *β* is the slope of the Weibull curve, which is the shape parameter of the Weibull distribution. The larger the *β* value, the smaller the Weibull dispersion. This indicates that the data points of the 2^#^ and 3^#^ samples are more scattered, and the randomness of the fatigue failure is also large. This is related to USRP treatment. Increasing the static load reduces the surface roughness and increases the hardness and residual stress; however, in addition to the roughness, the other two parameters are randomly increased on the material surface, which causes the local stress state of the material to be unstable during the fatigue process and in turn causes the fatigue life to be excessively dispersed.

When comparing the characteristic life *N*_a_ (as shown in Figure 5a–c), it is found that the life sequentially increases in the order of 0^#^, 2^#^, and then 3^#^. This is consistent with the expected effect. As the static load increases (the S value in Figure 5 corresponds to the stress value in GPa), the fatigue life increases. However, when the static load is too large, the surface properties of the material decrease. In contrast, a variety of fatigue crack sources are formed that are likely to cause fatigue cracking.

Subsequently, based on the bending fatigue life test data which were taken in Section 2.2 for large sample 20Cr2Ni4A, a more accurate life prediction curve, the *P*-*S*-*N* curve, is established to compensate for the limitations of the *P*-*N* curve. The relationship between bending stress *S* and fatigue life *N*, as shown in Equation (4), is used to establish the *P*-*S*-*N* life prediction curve of 20Cr2Ni4A material:(4)$$NS^{m}=C$$
where *N* is the bending fatigue life, *S* is the bending stress, and *m* and *C* are the parameters to be solved. Take the logarithm on both sides of Equation (4) to obtain Equation (5):(5)$$\ln S=-\frac{1}{m}\ln N+\frac{\ln C}{m}$$

The least squares method is used to fit the regression equation to determine the values of the parameters *m* and *C* to be sought. The specific formulas are shown in Equations (6) and (7):(6)$$-\frac{1}{m}=\frac{n\sum_{i=1}^{n}\ln N_{i}\times \ln S_{i}-\sum_{i=1}^{n}\ln N_{i}\times \sum_{i=1}^{n}\ln S_{i}}{n\sum_{i=1}^{n}{\left(\ln N_{i}\right)}^{2}-{\left(\sum_{i=1}^{n}\ln N_{i}\right)}^{2}}$$
(7)$$\frac{\ln C}{m}=\frac{\sum_{i=1}^{n}{\left(\ln N_{i}\right)}^{2}\times \ln S_{i}-\sum_{i=1}^{n}\ln N_{i}\times \sum_{i=1}^{n}\ln S_{i}\times \ln N_{i}}{n\sum_{i=1}^{n}{\left(\ln N_{i}\right)}^{2}-{\left(\sum_{i=1}^{n}\ln N_{i}\right)}^{2}}$$

If the values of *m* and *C* are required, the contact stress *S_i_* and fatigue life *N_i_* corresponding to different failure probabilities *P* need to be determined in conjunction with the Weibull distribution function. Based on the two-parameter Weibull distribution function, the bending fatigue life of the two process materials is calculated at three typical failure probabilities *P* (*P*_1_ = 10%, *P*_2_ = 50%, and *P*_3_ = 90%). The calculation results are substituted into Equations (6) and (7), and the test results of the three models are presented in Table 6:

The parameters *m* and *C* of the *P*-*S*-*N* curve in Table 6 are substituted into Equation (4), and the bending fatigue life *P*-*S*-*N* curves of the two process materials are obtained (as shown in Figure 6). According to the *P*-*S*-*N* curve, the bending fatigue life of the 20Cr2Ni4A material at different stress levels and the fatigue limit at any cycle can be obtained, and the reliability of the data can be improved according to the failure probability. From the perspective of reliable data, the bending fatigue limit of 20Cr2Ni4A steel at three million cycles under the lowest probability of failure (*P*_1_ = 10%) is also the highest probability of survival. The bending fatigue limits of the 0^#^, 2^#^ and 3^#^ samples are 651.36, 918.88, and 904.21 MPa, respectively.

### 3.2. Results of the Fatigue Fracture Morphologies

The fracture morphologies of the samples are shown in Figure 7. Figure 7a,b are the results for the sample 0^#^ after 50 and 500 magnification times. It can be seen that crack initiation mainly occurs in the surface defects, and there are obvious inclusions on the surface. It shows that the 0^#^ surface without USRP treatment has poor performance, and it is easy to form a fatigue crack source on the surface, which eventually leads to bending fatigue failure, and there is no obvious strengthening layer in Figure 7b. Figure 7c,d show the results of the fatigue fracture of sample 2^#^ after 50 and 500 times. The shape of the fatigue fracture surface on sample 2^#^ can be clearly divided. Moreover, the crack initiation site is not on the surface of the material but inside the material, and the final failure is caused by internal defects [24]. This is mainly because the surface properties of the 2^#^ sample are optimized by the USRP. The surface properties have been significantly improved. Therefore, it is difficult to give priority to the formation of crack sources on the surface under the same stress loading, but at the secondary surface or the internal defects of the material. The fatigue performance of sample 2^#^ is also better than that of sample 0^#^. Figure 7d shows that the strengthening layer is obvious, and its thickness is uniform. These USRP parameter settings are good, and its surface properties are the best. Figure 7e,f are the fatigue fractures of sample 3^#^. There are two crack initiation areas on the fracture for sample 3^#^, which are on the surface and inside of the material. This is related to the excessive static load parameter in the USRP, which leads to an uneven reinforcement layer, as shown in Figure 7f. The strengthening layer exists from 7.51 μm to 28.18 μm, the inclusion defects of the surface layer are also incorporated into the reinforcement layer, and then a large residual stress is introduced. The fatigue performance is relatively better than that for sample 0^#^ without treatment.

Figure 8 shows the internal results of the fatigue fractures of the three types of samples. As the internal structures of the three materials are consistent, they are introduced together. Figure 8a shows an intergranular fracture surface mainly composed of brittle fractures in the centre of the sprouting area. This area is repeatedly loaded during fatigue, resulting in the continuous crushing of both fracture sides. Figure 8b shows dimples, and there is a small section of the extrusion surface around the dimples, which is due to an excessive load during the fatigue process. The structure of the partial section is still in service because the overall structure does not fail after the fracture. Figure 8c shows an interface between the brittle fracture regions and dimples. The red marks in the figure indicate the direction of fracture propagation. Due to the difference in the two morphologies, a large crack is finally formed at the junction, which is consistent with the characteristics of structural parts in service conditions. Figure 8d shows typical fatigue striations distributed throughout the fracture, which reflects the process of fatigue loading.

## 4. Discussion

### 4.1. Mechanism of the Surface Properties on the Fatigue Performance

The concept of surface properties includes two aspects. Firstly, the surface quality of materials is usually characterized by surface roughness and surface microstructure. Secondly, there is the plastic deformation of the workpiece surface, residual stress and surface hardness. For different static load parameters, the surface integrity of the samples is different, and the influence on the bending fatigue performance is also different. On this basis, the influence of the surface properties of 20Cr2Ni4A steel on the fatigue performance was summarized from two aspects of surface morphology and surface mechanical properties.

#### 4.1.1. Effects of the Surface Mechanical Properties on the Fatigue Performance

In this section, surface hardness and surface residual stress are discussed. First, the surface microhardness of the samples under four static loads was measured by a microhardness tester. To ensure the accuracy of the test results, 15 test points were obtained for each group of static load parameters, and then the average values were taken. The results are shown in Figure 9.

Figure 9 shows that after the USRP treatment, the surface microhardness of the sample increases significantly. The surface microhardness of the sample before strengthening is 652 HV, and as the static load increases, the surface microhardness first increases and then decreases. When the peak occurs at 1374 N, the maximum average microhardness of the surface is 828 HV, which is 27% higher than that of the original sample. This is because with an increase in the static load, the plastic deformation causes the grains to slip and the grain boundary area to increase after the grains are refined. Additionally, the resistance of dislocation movement increases, and the dislocation density between grains increases. Thus, a work hardened layer is produced. The appearance of the work hardened layer increases the hardness of the sample surface. When the static load is too large, the resistance of the tool head also increases, which further increases the friction between the tool head and the surface of the test piece. The effect of frictional heat may be greater than that of cold plastic deformation. The material surface is softened and cold. The reduced hardness effect decreases the microhardness; at the same time, it affects the stability of the processing, resulting in non-uniform processing and flow of the material surface layer. Plastic deformation occurs and the local stress exceeds the tensile strength (such as deformation convexity). The grains at the beginning are not tightly arranged, which causes plastic deformation of the surface structure, and the dislocation structure is redistributed, leading to a decrease in the hardness. The bending fatigue performance is closely related to the surface hardness, and the fatigue limit is positively correlated with the change in hardness.

Then, the surface residual stress is introduced. During the USRP, the surface residual stress distribution obtained by different static load values is different. The residual stress results corresponding to the four types of static load values are shown in Figure 10.

Figure 10 shows that there is tensile stress on the sample surface under the USRP. This is because the tensile stress caused by the cutting force partially or completely offsets the residual compressive stress on the surface that is introduced by the carburizing process before the USRP. As a result, the residual compressive stress of the initial sample decreases or becomes tensile. After the USRP treatment, a large residual compressive stress is introduced on the sample surface. Under the same static load, the residual compressive stress on the sample surface is not much different. When the static load before the USRP treatment is 0, the average surface residual compressive stress is the smallest and is −29 MPa. As the static load increases, the residual stress tends to increase first and then decrease. When the static load is 1374 N, the average surface residual compressive stress is the largest and reaches −612 MPa.

The essence of residual stress is that a material undergoes lattice distortion under the action of an external force, and the crystal does not have the time to recover quickly and needs to maintain a microscopic force balance formed by the normal structure. Therefore, during material processing, residual compressive stress is often introduced on the structural parts to ensure that they remain stable during service, suppress the initiation and development of cracks, and improve the fatigue strength and corrosion resistance of the parts. This phenomenon can also be found by comparing the test results in Section 3.1. With the increase in static load pressure, the surface of the sample will produce more plastic deformation and introduce additional residual stress, which effectively improves the fatigue performance. However, when the static load pressure continues to increase, the local structure will undergo frictional heat and plastic deformation, which will lead to the release of local residual stress and a reduction in residual stress, and the fatigue performance will also decrease.

#### 4.1.2. Effects of the Surface Morphology on the Fatigue Performance

The relationship between surface states and fatigue performance is further analysed according to the test results in Figure 3, the surface roughness curves of the samples under four static loads are extracted, and three curves at equal intervals under each static load parameter are measured. The value is taken as the average surface roughness under this parameter, and the results are presented in Table 7.

Table 7 shows that the average surface roughness of sample 0^#^ is the largest, and that for sample 2^#^ is the smallest. As the static load increases, the average surface roughness decreases first and then increases. When the static load is 1374 N, the minimum average surface roughness is 0.114 μm, which is approximately five times lower than that before strengthening, indicating a substantial reduction in the roughness of the material surface. When the static load is 1963 N, the static load is very large, which causes the surface plastic deformation to be too large, the surface roughness to increase, and the average surface roughness to slightly increase to 0.197 μm.

In summary, the milling marks on the front surface after the USRP treatment are very obvious, and the “peaks and valleys” are staggered. After the USRP, there is no obvious difference between the peaks and valleys, and the surface roughness is reduced. This is because the punch presses the “peak” into the “trough”, the “trough” is filled with the material from the “peak”, and the difference between the peaks and valleys is reduced. As the static load increases, the surface roughness of the sample further decreases. With an increase in the static load and due to the effect of the cooling liquid, the surface temperature rapidly drops, and the work hardening effect becomes very obvious. The friction force generated by the USRP is 10–30% that of the traditional rolling process, and the friction is relatively small, so the surface roughness of the material is reduced, which can improve the surface quality. However, if the static load is too high, the anti-wear effect of the ultrasonic tool head will be weakened, causing new surface roughness, destroying surface roughness, increasing surface roughness, and reducing surface quality. Compared with the results of fatigue limit, it is found that the fatigue performance is positively correlated with the surface roughness.

Subsequently, the morphology of the samples before and after USRP treatment was observed by TEM, and the results are shown in Figure 11. Figure 11a shows sample 0^#^, and Figure 11b–d show the results for sample 2^#^. It can be seen from the figure that after the USRP treatment, the grain refinement and grain boundaries in the material increase due to the slippage, proliferation and entangling of the dislocations, which are caused by plastic deformation.

As shown in Figure 11a, the surface layer before USRP treatment is mainly composed of high-carbon martensite, and its substructure contains martensite twins. There are no obvious dislocations and large-angle cross-entanglement, and no obvious dislocation accumulation. Figure 11b shows that the dislocation density increases significantly after USRP treatment, and a large number of dislocations exist in some grains. It can be seen in Figure 11c,d that after USRP treatment, dislocation slip on the surface is obvious, and the dislocations continue to move inside the crystal grains, resulting in dislocation segment plugging. The dislocation movement must overcome point defects, foreign body atoms, defect groups, and stress fields. If these obstacles are not passed, the dislocations stop moving and eventually form a dislocation plug [25]. Fine-grain strengthening during USRP treatment occurs because of an increase in the number of dislocations. The microscopic mechanism is that the material surface undergoes strong plastic deformation, causing dislocations to slip, multiply and accumulate as entanglements, forming dislocation walls. With a further increase in the strain and strain rate, the dislocation wall gradually transforms into a sub-grain structure, and the sub-grain structure eventually transforms into a high-angle grain boundary, thereby causing grain refinement and improving the fatigue strength of the material.

XRD spectra are then obtained to analyse the dislocation density of the samples before and after USRP treatment, and the results are shown in Figure 12. The XRD diffraction peak positions do not change significantly before and after the treatment, indicating that no new phase was formed. The main reason for the broadening of the diffraction peak full width at half maximum (FWHM) is the refinement of grains and the existence of residual stress. Residual stress is caused by an uneven volume change and plastic deformation of the metal internal structure, which is essentially a lattice distortion. However, lattice distortion and grain refinement are mainly caused by dislocations, so the root cause is the change in dislocation density. To characterize the change in the dislocation density of 20Cr2Ni4A carburized steel before and after USRP treatment, the FWHM of the diffraction peaks corresponding to the (110), (200), and (211) crystal planes is used for calculation.

The dislocation density can be determined by the Dunn [26] formula, as shown in Equation (8):(8)$$D=\frac{\beta^{2}}{2\ln2\pi \times b^{2}}=\frac{\beta^{2}}{4.35b^{2}}$$
where *D* represents the dislocation density, *b* represents the Burgers vector, and *β* represents the FWHM. The integral method can be used to make the tangent line L at the bottom of the peak.

As all samples are made of the same material, the Burgers vector b can be regarded as a constant, and the dislocation density *D* is proportional to the half-maximum width *β*/2. Therefore, the dislocation density on the sample surface can be analysed by XRD.

It can be seen that with an increase in the *β* value of the X-ray diffraction peaks, the dislocation density in the materials treated with the USRP increases. The FWHM of the diffraction peaks corresponding to the three crystal planes in Figure 12 is measured and compared using Equation (8). The diffraction peak FWHM before and after the treatment is presented in Table 8.

In Table 8 and Figure 12, it can be seen that the dislocation density increased significantly after the USRP treatment. The relative change rate of the dislocation density based on the (200) diffraction peak was 92.51% at the maximum and 7.62% at the minimum. The dislocation density represented by the diffraction peaks increases to varying degrees, and the relative change rates of the dislocation densities corresponding to the b and c diffraction peaks are significantly different, indicating that the dislocations mainly slip along the (200) and (211) crystal planes, and the direction change along (110) is not obvious. The final result shows that the dislocation density of the samples treated with the USRP has significantly increased, and it is further verified from the microstructure that the USRP treatment changes the material at the microscopic level, which increases the dislocation density of the material and further increases the residual stress on the material surface. Microhardness and grain refinement ultimately improve the bending fatigue properties of the material.

### 4.2. Mechanism of the Cross-Sectional Structure on Fatigue Performance

This section analyses the mechanism of the change in the cross-sectional structure of the samples. First, based on the gradient change of the residual stresses and microhardness values of the sample cross-sections, the fatigue performance improvement after USRP treatment in different procedures is analysed. Based on the EBSD test results and metallographic structure test results, the phase structure of the samples treated with the USRP is analysed, and the bending fatigue test data in Section 3.1 are used to analyse the mechanism of the phase structure in the fatigue process.

#### 4.2.1. Effects of the Residual Stress on the Fatigue Performance

During the USRP treatment, the sample surface is extruded by the high-frequency impact. The sample surface undergoes large-scale plastic deformation, which leads to lattice distortion and increases the number of dislocations. The material surface mainly undergoes plastic deformation, while the second surface mainly undergoes elastic deformation. When unloading, the secondary surface rebounds and cannot be recovered due to the obstacle of permanent plastic deformation of the surface. The surface becomes compressed, and the second surface is in a state of stress, which results in a residual stress over a certain depth range. The peeling method [27] is used to measure the residual stress value of the sample in the depth direction before and after the USRP treatment, and the measurement results are shown in Figure 13.

It can be seen in Figure 13 that the residual compressive stress layer on the surface layer of the carburized original sample has a certain depth. This is because the carbon content of the carburized layer gradually decreases from the outside to the inside. The higher the carbon content, the larger the martensite transformation. When the temperature is low, the martensitic phase transition first occurs at the junction between the cementation layer and the matrix and then changes in both directions, with the high-carbon surface finally changing. The martensitic transformation is accompanied by an increase in volume. The surface layer of the last transformation is constrained by the heat that completed the first transformation and is subject to expansion to form a residual compressive stress layer [28]. After the USRP treatment, a high value residual compressive stress layer is introduced into the sample surface. The thickness of the layer exceeds 1200 μm. The generated residual compressive stress is distributed along with the depth of the layer, and as the depth increases, the residual compressive stress first increases and then decreases. The final stress level is similar to that of the untreated sample. The maximum residual compressive stress value reaches 1359 MPa, which is located 350 μm from the surface and even exceeds the yield strength of the 20Cr2Ni4A carburized steel. This is because the USRP treatment strengthens the material, and its yield strength also increases. The reason for the peak value of the residual compressive stress at 0.3–0.4mm from the surface is that the surface is cut during the samples test. The material undergoes yield deformation from the surface and inside. Part of the residual stress is released during the deformation process. However, with an accumulation of energy and the layer-by-layer transfer, the residual stress continues to increase, which represents a change in the internal residual stress caused by the USRP treatment. Comparing the test results in Section 3.2, the bending fatigue limits of the 0^#^, 2^#^ and 3^#^ samples are 996.34, 1085.51 and 1073.19 MPa, respectively. It can be seen that the USRP treatment increases the material residual stress, and the existence of compressive stress and gradient in the residual compressive stress hinders the propagation of fatigue cracks and improves the fatigue strength of the material. However, excessive USRP treatment causes surface defects, a residual stress release, and an uneven distribution in the USRP treatment layer. This easily causes stress concentrations when subjected to bending loads that eventually form a fatigue source and decrease the fatigue life of the material.

#### 4.2.2. Effects of the Microhardness on the Fatigue Performance

While introducing a layer of residual compressive stress, the USRP also increases the hardness of the material surface layer to a certain extent, which can be attributed to the combined effect of grain refinement and work hardening. The distribution of microhardness values along the depth direction of samples before and after USRP treatment is measured, and the results are shown in Figure 14.

It can be seen in Figure 14 that the microhardness of the three samples before and after USRP treatment has a gradient. As the layer depth increases, the microhardness value increases first and then decreases and finally gradually decreases to the matrix material hardness. After treatment, the microhardness values of the 2^#^ and 3^#^ materials increase significantly within a certain range. The thickness of the layer with an increased microhardness can reach more than 1000 μm and then reach the microhardness level before treatment. The reason is that with increasing depth, the plastic deformation degree is smaller, the grain size is larger and the residual austenite content is larger; that is, the effect of work hardening, fine grain strengthening and phase structure strengthening decreases layer by layer. The area closest to the surface is closer to the boundary, which is less constrained and more prone to deformation. At the same time, after the treatment, the surface of the sample recovers elastically, the density between the grains is reduced, and the maximum microhardness value never occurs. On the subsurface, this is consistent with the measured residual compressive stress distribution. The maximum microhardness value of sample 3^#^ is 802 HV at a distance of 150 μm from the surface. This is because sample 3^#^ undergoes an excessive USRP treatment, and the hardness increase is very obvious. The 2^#^ sample is significantly improved, but it is 35 HV smaller than that of sample 3^#^.

At the same time, along with the residual compressive stress affecting the layer, it can be seen that the thickness of the layer affected by the mechanical properties of USRP is larger than that of the grain refined layer, which shows that the area affected by the USRP is not limited to the grain refinement layer. After blocking layer by layer, the energy is attenuated to the subsurface. Significant changes have occurred in the microstructure, but the crystal grains in this area still have a certain lattice distortion, but the surface has been strengthened. The microhardness changes along the depth of the layer inhibit fatigue crack growth. The results of the microhardness and the grain size and residual compressive stress also explain the reason for the narrowing of the fatigue striations gap after treatment from a micro perspective.

#### 4.2.3. Effects of the Phase Structure on the Fatigue Performance

The changes in both the residual stress and microhardness have a significant effect on the fatigue performance, but the mechanism of these two parameters is related to the phase structure in the sample. Therefore, if we want to analyse the mechanism that controls the fatigue performance of materials that undergo the USRP, the phase structure before and after the USRP treatment must be observed. As the USRP parameters for sample 3^#^ are too large, to accurately analyse the impact of the USRP treatment on fatigue performance, only samples 0^#^ and 2^#^ are selected for EBSD (Electron Backscattered Diffraction) analysis. The results are shown in Figure 15.

It can be seen in Figure 15 that the USRP significantly reduces the grain size on the sample surface. The average grain size on the surface before treatment is 2.130 μm (as shown in Figure 15a), and it decreases to 0.674 μm (as shown in Figure 15b) after USRP treatment. The surface grain orientation distribution before and after USRP treatment is shown in Figure 15c,d. The black lines in the figure indicate the high-angle grain boundaries with orientation differences greater than 15°, the red lines represent low-angle grain boundaries with orientation differences less than 15°, and the white part is the matrix structure. The difference in orientation between adjacent grains also affects crack propagation and material fracture to an extent. As the grain orientation difference increases, the interfacial energy increases, and the grain boundary energy barrier rises, which hinders crack growth. At the same time, the high-angle grain boundaries have a greater impact on the crack propagation path. As the number of high-angle grain boundaries increases, the effect of suppressing crack growth is enhanced [29].

It can be seen that after the USRP treatment, in addition to the obvious changes in the grain size and orientation of the tested sample surface, the residual austenite content on the surface is significantly reduced. The results are shown in Figure 15e,f. The blue regions indicate the presence of austenite. Upon comparing Figure 15e,f, it is clear that after the USRP treatment, the number and area of the blue regions in the figure are significantly reduced. The relative content is 0.56%, which is calculated based on the area occupied by the blue parts.

In summary, the surface undergoes severe plastic deformation after surface strengthening, which significantly refines the original martensitic grains, reduces the retained austenite, breaks the grains, and increases the number of grain boundaries, which can hinder fatigue crack growth. Since plastic deformation is a process of energy accumulation and layer-by-layer transfer, the grain size gradually increases from the surface layer to the interior of the material. According to this situation, the grain structure of the section position of the sample treated by USRP was analysed, and the results are shown in Figure 16, where Figure 16a is the surface location of the sample, and Figure 16b–d are 10 μm, 500 μm and 10,000 μm from the surface, respectively. Based on the analysis of the EBSD phase structure and the changes in the gradient in the residual stress and microhardness, a cross-sectional microstructure schematic diagram of the 20Cr2Ni4A carburized steel sample subjected to the USRP standard treatment process is shown in Figure 16. The grain diagram refers to the research results of Zhao [30], and there are five layers from the surface to the core, which are symmetrically distributed.

The first layer is a USRP-reinforced layer (0–8 μm), which shows that the grains are sufficiently refined and distributed much more evenly. The residual compressive stress is released due to the cross-section incision, and it occurs from small to large. This layer belongs to the surface layer, its performance is the best, and the anti-fatigue performance is mostly related to the performance of this layer.

The second layer is a USRP-treated hardness release layer. The hardness of this layer decreases from large to small due to the USRP-treated surface, and it is finally released. It is worth noting that the layer has a lamellar phase structure in the initial stage. This structure is severely stretched due to internal compression after USRP treatment.

The third layer is a carburized layer. This layer is an edge position that can be achieved by carburizing. The phase structure of the material gradually approaches the core structure. However, due to the influence of the carburizing process, fine carbon particles exist in this layer, causing an increase in the hardness.

The fourth layer is the residual stress release layer caused by the USRP treatment. The reason a large amount of residual stress is released after cutting the material cross section is that there is a gradual reduction in the residual stress of the first three layers. In the fourth layer, the residual stress increased by the USRP treatment is also gradually released, and the grains continue to the core.

The last layer is the core of the material. This layer acts as a control layer and is not subjected to surface treatment. The phase structure, microhardness and residual stress of this layer do not change.

## 5. Conclusions

This paper focuses on the effect of USRP treatment on the bending fatigue properties of materials. Based on four sets of increasing static load control experiments, the mechanism of the USRP treatment on the materials is studied. The main research conclusions are as follows:(1)The 20Cr2Ni4A carburized gear steel herein has a high strain rate behaviour under the action of USRP. This behaviour significantly improves the bending fatigue performance of the steel. When the static load is 1374 N, the minimum surface roughness is 0.114 μm, and the maximum microhardness value and residual compressive stress value are 828 HV and 612 MPa, respectively. Compared with the samples without the USRP treatment, the surface roughness decreases by approximately 5 times, the surface microhardness increases by 27%, and the residual compressive stress on the surface increases by approximately 20 times. The fatigue strength of the material increases with decreasing surface roughness and increasing microhardness and residual compressive stress, and the optimal static load value is 1374 N.(2)The bending fatigue life of 20Cr2Ni4A carburized gear steel at 0, 1374, and 1963 N static load was tested, and the bending fatigue limit of the three samples was calculated by P-S-N curve fitting. The bending fatigue limits of samples 0^#^, 2^#^, and 3^#^ are 651.36, 918.88, and 904.21 MPa, respectively. The fatigue fracture was analysed using SEM. It was found that the fatigue source in sample 0^#^ without the USRP treatment is on the sample surface; the fatigue source in sample 2^#^ after the standard USRP treatment is mostly on the subsurface or inside the sample during the fatigue process, and the gap between the fatigue stripes decreases. The fatigue source in sample 3^#^ with a large static load is generated unevenly on the surface, and the crack source is not singular in nature.(3)According to the sample surface structure analysis of the influence of USRP treatment on the fatigue performance, USRP treatment significantly improved the surface roughness, increased surface hardness, introduced a lot of residual compressive stress, introduced a large number of dislocation multiplications, and further improved the fatigue performance. It was also found that excessive USRP treatment would lead to increased surface roughness. Although a large amount of residual compressive stress was introduced and surface hardness was improved, the fatigue performance was still reduced.(4)Combined with EBSD phase structure analysis and the change in gradient residual stress and microhardness, the section microstructure diagram of 20Cr2Ni4A steel after the USRP standard treatment process is drawn. The grain diagram is divided into five layers from the surface layer to the core, showing a symmetrical distribution. The grain size, residual stress and hardness of the gradient distribution inhibit the initiation and propagation of fatigue cracks, which is also the reason why the crack source takes place on the subsurface and the fatigue striation spacing decreases.

## Figures and Tables

**Figure 1 materials-14-02516-f001:**
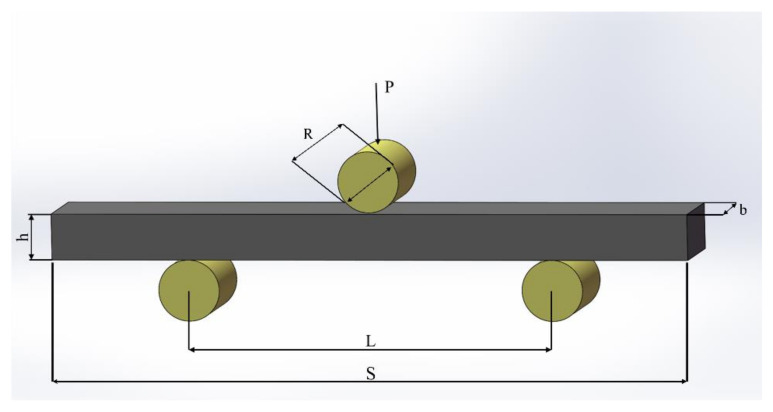
Schematic diagram of the sample size.

**Figure 2 materials-14-02516-f002:**
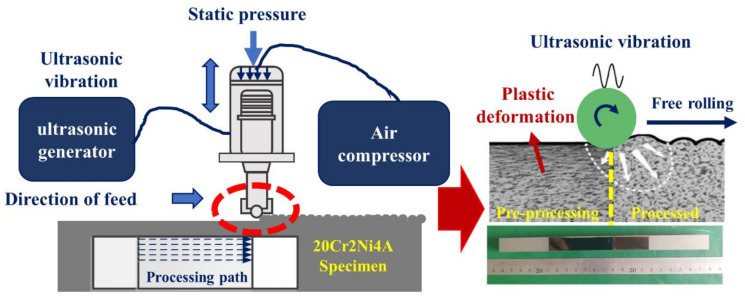
USRP equipment schematic.

**Figure 3 materials-14-02516-f003:**
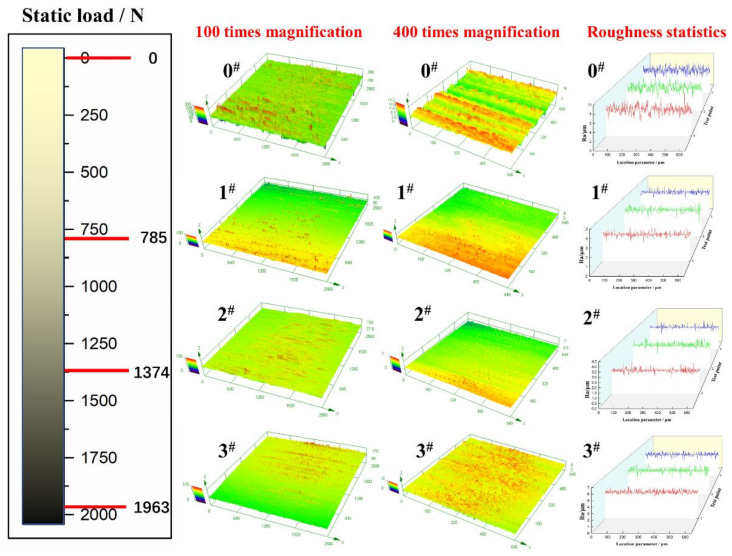
Three-dimensional topography before and after processing.

**Figure 4 materials-14-02516-f004:**
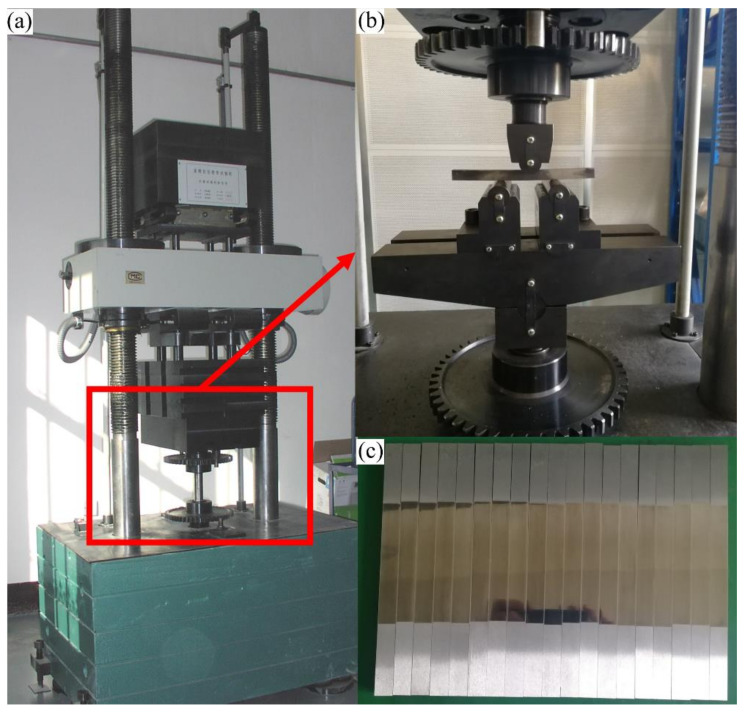
Three-point bending fatigue test device. (**a**) Testing machine, (**b**) three-point bending fixture, (**c**) samples treated with URSP.

**Figure 5 materials-14-02516-f005:**
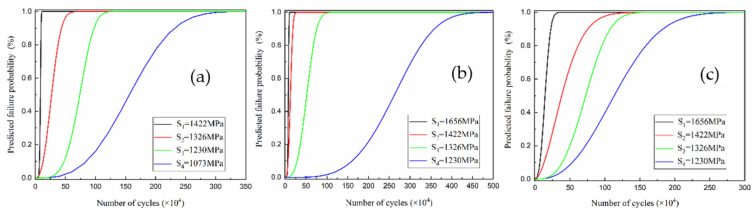
P-N curves of the three-point bending fatigue tests before and after USRP treatment. (**a**) Sample 0^#^, (**b**) sample 2^#^, and (**c**) sample 3^#^.

**Figure 6 materials-14-02516-f006:**
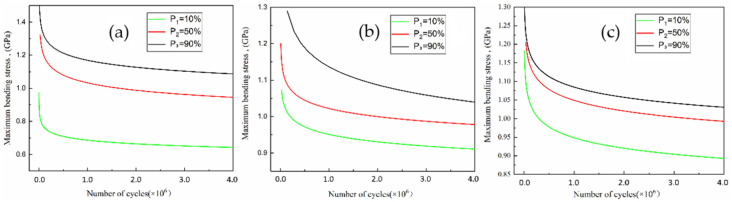
*P-S-N* life curves from the three-point bending fatigue tests for (**a**) sample 0^#^, (**b**) sample 2^#^, and (**c**) sample 3^#^.

**Figure 7 materials-14-02516-f007:**
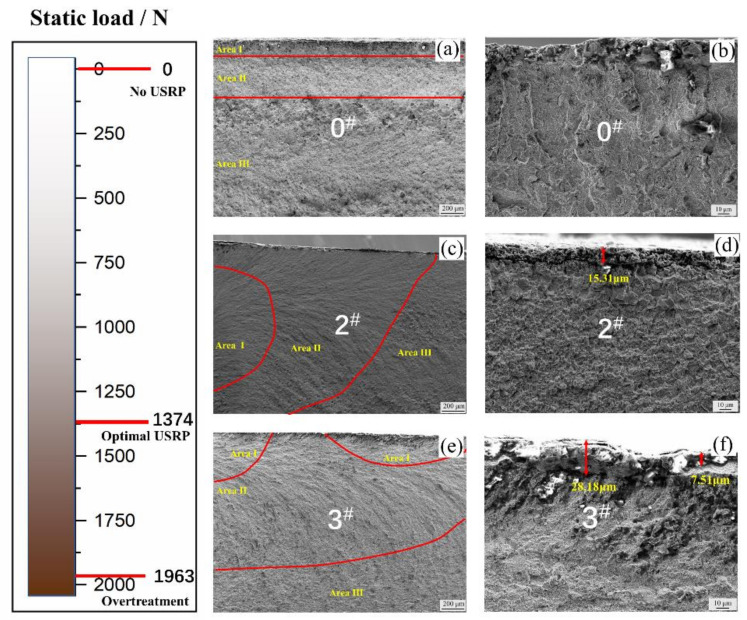
SEM images of fatigue fractures for sample (**a**) 0^#^ at 50× magnification, (**b**) 0^#^ at 500× magnification, (**c**) 2^#^ at 50× magnification, (**d**) 2^#^ at 500× magnification, (**e**) 3^#^ at 50× magnification, (**f**) 3^#^ at 500× magnification.

**Figure 8 materials-14-02516-f008:**
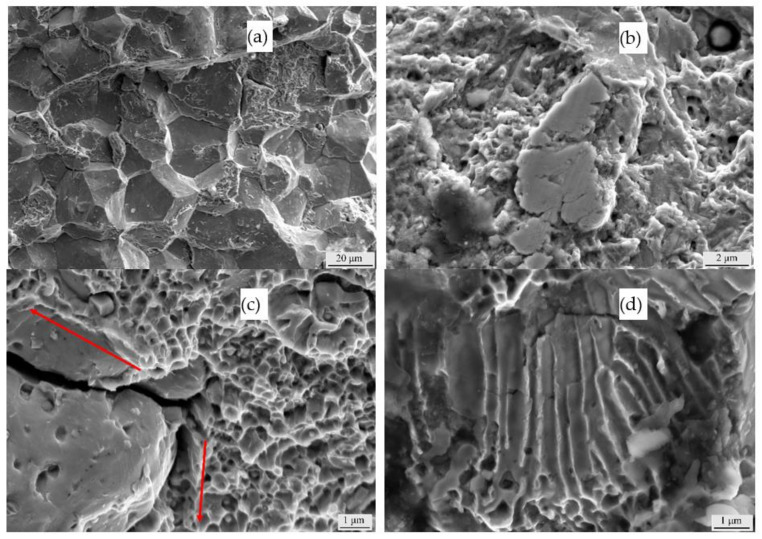
SEM images of fatigue fractures after USRP: (**a**) river pattern, (**b**) extrusion surface, (**c**) crack morphology, and (**d**) fatigue pattern.

**Figure 9 materials-14-02516-f009:**
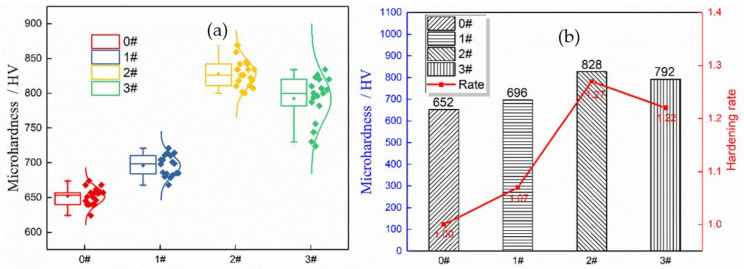
Microhardness value of the sample: (**a**) box pattern, (**b**) mean values of the microhardness.

**Figure 10 materials-14-02516-f010:**
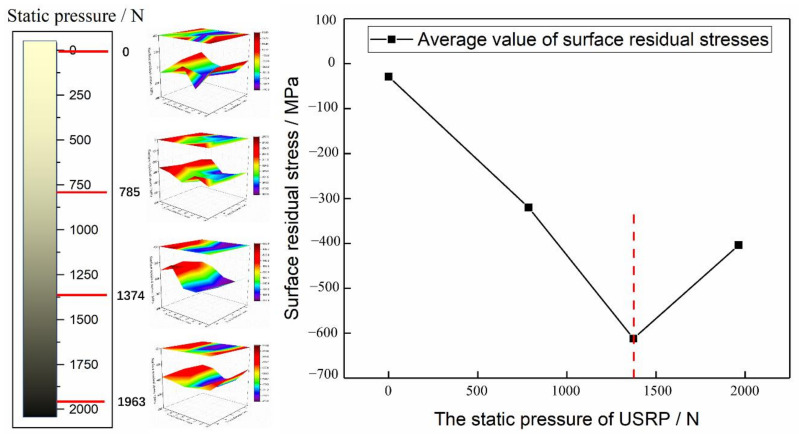
Effect of static load on the surface residual stress.

**Figure 11 materials-14-02516-f011:**
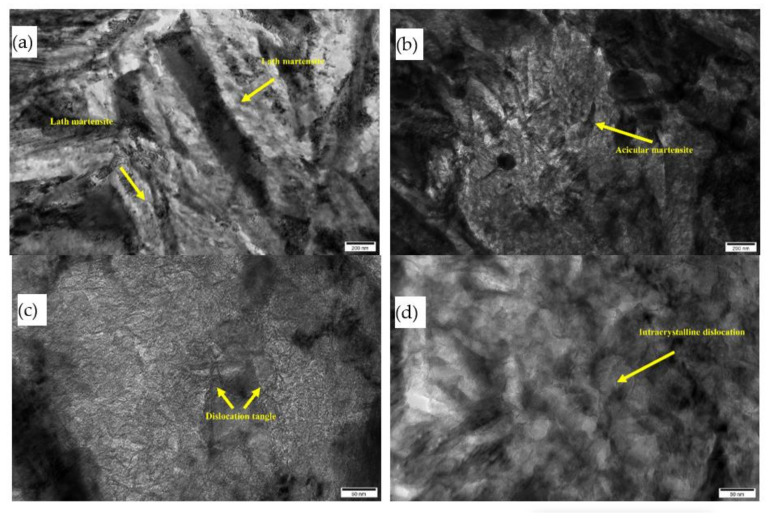
TEM images before and after USRP: (**a**) 0^#^ sample surface micromorphology, (**b**) 2^#^ sample surface micromorphology, (**c**) 2^#^ sample intercrystalline dislocations, (**d**) 2^#^ intercrystalline dislocations.

**Figure 12 materials-14-02516-f012:**
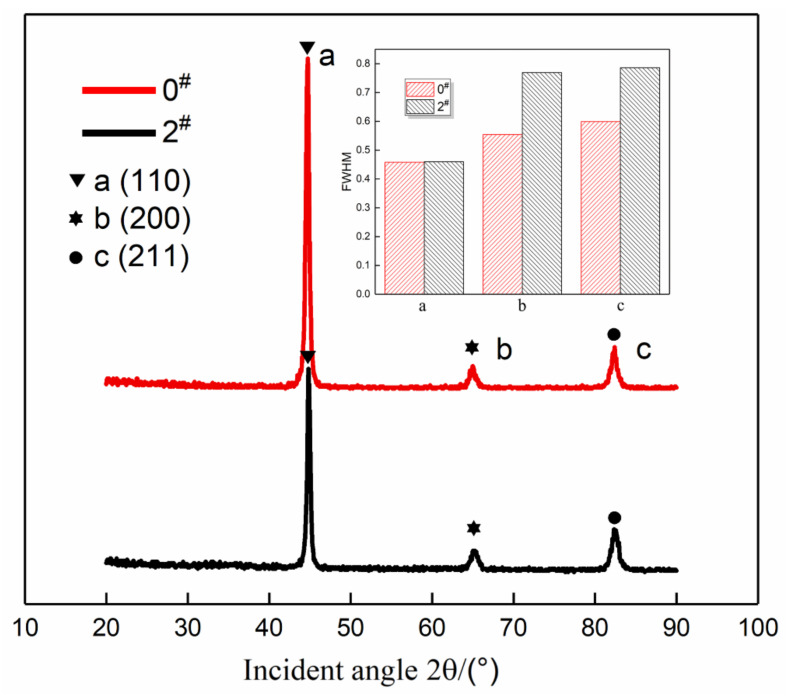
XRD spectra before and after the USRP treatment.

**Figure 13 materials-14-02516-f013:**
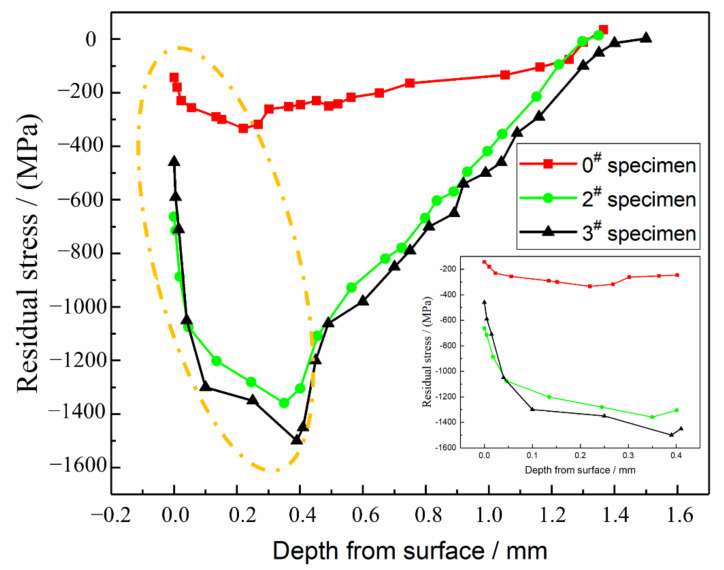
Residual stress along with the depth of the layer before and after the USRP treatment.

**Figure 14 materials-14-02516-f014:**
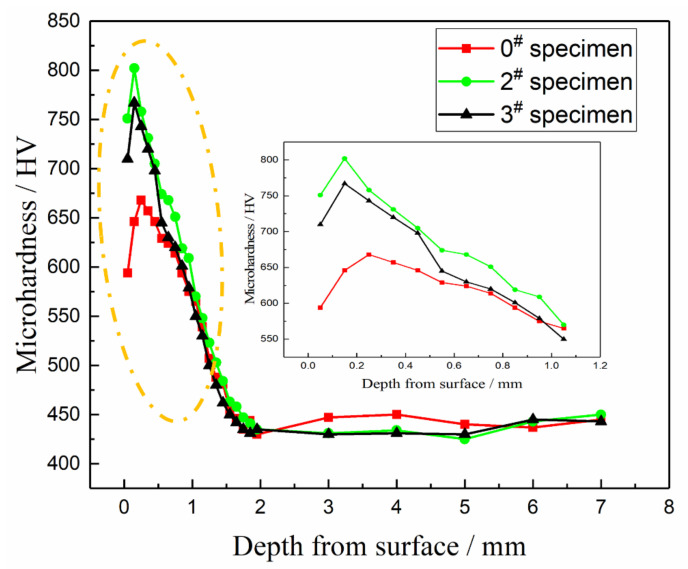
Microhardness along the depth direction before and after the USRP treatment.

**Figure 15 materials-14-02516-f015:**
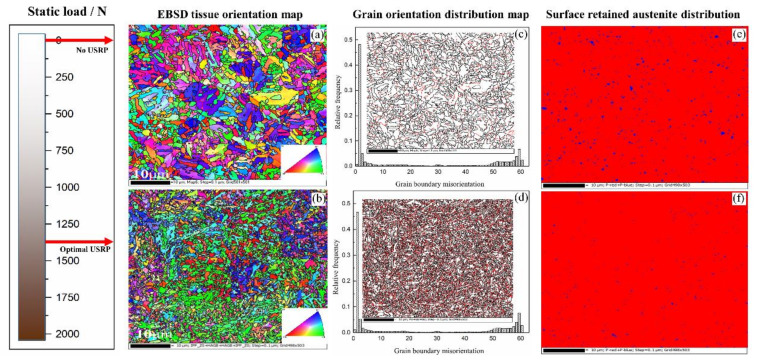
EBSD graphical results before and after the USRP treatment: (**a**) EBSD tissue orientation of sample 0^#^, (**b**) EBSD tissue orientation of sample 2^#^, (**c**) grain orientation distribution of sample 0^#^, (**d**) grain orientation distribution of sample 2^#^, (**e**) surface retained austenite distribution of sample 0^#^, (**f**) surface retained austenite distribution of sample 2^#^.

**Figure 16 materials-14-02516-f016:**
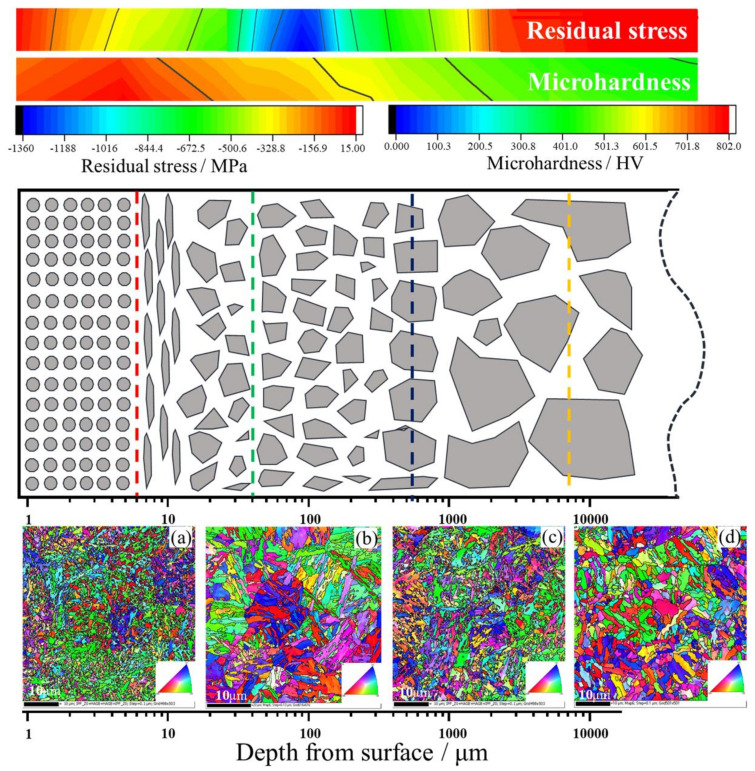
Schematic diagram of the microstructure of the carburized steel after USRP treatment: (**a**) EBSD tissue orientation of the surface, (**b**) 10 microns from the surface, (**c**) 500 microns from the surface, (**d**) 10,000 microns from the surface.

**Table 1 materials-14-02516-t001:** Chemical composition and content of the 20Cr2Ni4A steel (wt.%).

Element	Cr	Ni	Mn	Si	Al	S	O
Percentage composition	1.25–1.65	3.25–3.65	0.30–0.60	0.15–0.35	≤0.01	≤0.005	≤0.0012

**Table 2 materials-14-02516-t002:** Temperatures used for the heat treatment process of the 20Cr2Ni4 gear steel.

Heat Treatment Process				
Normalizing Temperature/°C	Carburizing Temperature/°C	High-Temperature Tempering/°C	Quenching Temperature/°C	Low-Temperature Tempering/°C
950	920	640	800	150

**Table 3 materials-14-02516-t003:** Process parameters for the USRP treatment.

Sample Number	Line Speed m/min	Step mm	Amplitude μm	Static Load N
0^#^	0	0	0	0
1^#^	2	0.08	20	785
2^#^	2	0.08	20	1374
3^#^	2	0.08	20	1963

**Table 4 materials-14-02516-t004:** Three-point bending fatigue test plan.

Number	Mean Load (kN)	Alternating Load (kN)	Test Frequency (Hz)	Bending Stress Value (MPa)
1	35	11.6	79	1656
2	30	10	79	1422
3	28	9.3	79	1326
4	26	8.6	79	1230

**Table 5 materials-14-02516-t005:** Weibull distribution function before and after the USRP treatment under different stress levels.

	*σ*_max_ (MPa)	*β*	*N* _a_	Fatigue Life Equivalent Value (10^4^ Cycle)		
				*P*_1_ = 10%	*P*_2_ = 50%	*P*_3_ = 90%
0^#^	1422	9.12	88,187	6.88	8.47	9.66
	1326	2.54	312,447	12.87	27.04	43.41
	1230	4.34	804,220	47.88	73.91	97.46
	1073	3.07	1,764,083	84.82	156.57	231.41
2^#^	1656	7.01	78,900	5.79	7.57	8.99
	1422	3.09	141,436	6.83	12.56	18.53
	1326	3.15	581,996	28.45	51.80	75.87
	1230	3.84	2,886,819	160.70	262.41	358.68
3^#^	1656	2.71	164,714	7.18	14.38	22.41
	1422	1.75	469,887	13.02	38.12	75.61
	1326	3.03	817,778	38.93	72.47	107.67
	1230	2.50	1,311.653	53.28	113.26	183.16

**Table 6 materials-14-02516-t006:** Parameter *m* and *C* values of the *P-S-N* curve under different failure probabilities.

Sample	Failure Probability (%)	*m*	*C*
0^#^	10	9.99	0.0296
	50	10.87	0.3323
	90	11.72	0.0643
2^#^	10	13.92	0.0096
	50	13.69	0.1989
	90	13.69	0.7301
3^#^	10	7.69	0.0006
	50	7.08	0.2241
	90	7.06	1.4039

**Table 7 materials-14-02516-t007:** Surface roughness measured under different static loads.

Sample	Surface Roughness Ra/μm			Average/μm
	Area 1	Area 2	Area 3	
0^#^	0.626	0.686	0.602	0.638
1^#^	0.118	0.115	0.130	0.121
2^#^	0.107	0.146	0.090	0.114
3^#^	0.185	0.233	0.173	0.197

**Table 8 materials-14-02516-t008:** FWHM of diffraction peaks before and after USRP.

	Incident Angle (2*θ*/°)	a	b	c
		44.780	65.226	82.400
0^#^	*β* _1_	0.458	0.554	0.599
	*β* _1_ ^2^	0.210	0.307	0.359
2^#^	*β* _2_	0.460	0.769	0.786
	*β* _2_ ^2^	0.212	0.591	0.618
Relative variation ratio	Δ = 100% × (*β*_2_^2^ − *β*_1_^2^)/*β*_1_^2^	7.62%	92.51%	72.14%

## Data Availability

The data presented in this study are available on request from the corresponding author.

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
