# Peer review of "Bending Fatigue Behaviour and Fatigue Endurance Limit Prediction of 20Cr2Ni4A Gear Steel after the Ultrasonic Surface Rolling Process"

_materials, 2021, doi:10.3390/ma14102516_

Round 1

Reviewer 1 Report

The paper provides lot of experimental data and thus it is very interesting.

Anyway, there are lot of (probably) typing errors in all parts of text, including abstract. Please also check used units for consistency (mpa, Mpa,...) and English style (meaning of  text is not clear in some places).

It would be nice if it will be possible to make the Discussion shorter and more concentrated on the most important findings..

Anyway, I recommend this paper for publication.

Author Response

  1. For English language problems, we decided to conduct a detailed vocabulary and grammar check, and submit professional polishing at the end.
  2. In view of the excessive content of the Discussion, we simplified the content by modifying the framework of the Discussion part. The detailed modifications have been shown in the uploaded attachment.

Reviewer 2 Report

This paper has serious flaws which should be improved before the consideration for publication.

  1. The manuscript is slappy-written and some work is required to fix that matter. I will give only examples from the abstract:

Line 18: „mpa”, „Mpa”

Line 19: „inte-rior”

Line 20: „mate-rial”

Line 28: „sec-tion”.

  1. Information from lines 119-120 should be given in the table. Lines 126-127 should be replaced with a scheme or the dimensions placed in Fig. 1…
  2. Some language-issues also can be found, such as “microstructure generated by” (line 40) or “material organization continuity and stress ordering” (line 49). The authors should carefully check the entire manuscript.
  3. The introduction part is too generic and contains a very low number of references. There is a lot of techniques improving the fatigue performance by surface-treatment (e.g. shot peening) so why the Authors choose USRP instead? The only point which aims to highlight this issue is lines 88-89. This requires an explanation in that part of the manuscript. Additionally, some parts of the introduction are kind of “text-filler”, e.g. in lines 73-77 there is a discovery that cyclic loading in LCF regime causes stress relaxation… this is an obvious fact for every cycle in LCF causes small plastic deformation and it has not anything with USRP… Another thing in lines 57-61 “(…) the structure of the metal organization becomes denser, which improves the fatigue performance (…)”….? What is “metal organization”? What is the relation between density and fatigue performance? Is this really a phenomenon that improves the fatigue strength in the surface-treatment….? I do research in fatigue and I am not convinced at all….
  4. Fractography. Lines 331-335 “(…) crack initiation site is not on the surface of the material but inside the material (…) This is mainly because the surface performance (…) is optimized by the USRP”. I recommend the Authors analyze a state of stress on a surface and under it. Maybe it will give some explanation why the crack initiation is in that specific area and how USRP affects this phenomenon.
  5. Fractography again. Please, show me the “river pattern” on Fig. 8a with an arrow.. I only see an intergranular fracture surface. Fig. 8b: It does not look like an extrusion… Is it not just a sheared material? Fig 8d: are these fatigue striations taken from the same spot as striations from fig. 8b (right to “the sheared material” area)?

Author Response

  1. For English language problems, we decided to conduct a detailed vocabulary and grammar check, and submit professional polishing at the end.
  2. For the problem of unclear expressions in lines 119-120 and 126-127, we have added relevant parameters and modified Figure 1 as required
  3. In view of the general description of the introduction, we have consulted a large number of literatures as required, and rewrote the introduction. The introduction highlights the advantages of USRP, deleted the text filling content of lines 73-77, and modified the expression of lines 57-61 to change the fatigue performance into strength.
  4. For the problem that the origin of crack initiation cannot be explained clearly, we fully consider the load factor of the sample, and make a comprehensive expression with this factor.
  5. Regarding the problem of fatigue fracture description, we remove the expression of river pattern, and believe that there is still a process of compressive pressure to interfere with the shear force in the fatigue process, and Fig. 8d shows the fatigue striations distributed around the fracture.
  6. The detailed modifications have been shown in the uploaded attachment.

Reviewer 3 Report

The paper is fine because it is clear, the employed tools are the right ones and reaches a specific recipe to process this steel for gears.

The description of figure 5 is upside down. It must be corrected.

Author Response

  1. For the problems of the introduction, we have consulted a large number of literatures as required, and rewrote the introduction. Other surface reinforcement methods are listed and the advantages of USRP are highlighted.
  2. In line 120, the meaning of each parameter has been listed in detail, and introduced in millimeters.
  3. Figure 1 has been revised as required, with length b prominently placed and the diameter of the drum described.
  4. Fig. 2 is further described, the feed motion is on the tool, and the way of preprocessing is checked, and the unclear problem is corrected.
  5. For the problem of static pressure expression in this paper. All ‘static pressure’ in this article are modified to ‘static load’.
  6. The description of the static pressure correlation is simplified, and the high resolution results of Fig. 9 are given in the attachment
  7. In view of the hardness test instrument description is not clear, we introduced the hardness tester specifications and part of the experimental parameters in detail, in which the indenter of the durometer is a diamond cone indenter, and the pressure applied to the sample was 300 g.
  8. In view of the problem that the method of residual stress detection is not clearly described, the parameters of the instrument and the experimental parameters are discussed in detail, and the method in the literature (https://doi.org/10.1080/10910340903451472) has been explained in the paper.
  9. To solve the problem that the original fatigue data is not provided, we decide to provide the original fatigue test data in the attachment, and provide the Matlab calculation code to verify the reliability of the data in the paper.
  10. In view of the unclear discussion in the discussion part, Section 4.1 is rearranged, with emphasis on the influence of surface morphology and surface mechanical properties on fatigue performance.
  11. The TEM sampling position is the microscopic appearance of the sur-face layer treated by USRP.
  12. Burgers vector isadopted in the text.
  13. The maximum residual stress is the maximum compressive stress of the residual stress.

Reviewer 4 Report

In general, I think that this paper should be rewrote. The main problem is the communicative structure. There are a lot of information and analysis not well organized in order to converge on the focus of the paper. Results are presented without reports a clear discussion focalized on the objective of the research. These results are usually pertinent also with a specific appropriate discussion (about the result in itself) but the relation with the focus of the paper is weak.

  • Section 1: Introduction

The research proposed by the paper at end of the introduction (pg. 3, line 106) seems to be described too generically. The activities that cover the highlighted literature lacks could be listed.

  • Sections 2: materials and methods

Line 120 the meaning of symbols should be described e.g. meaning (symbol) XX%.

Line 120 if you write S =14 h all sizes are a function of the same parameter h and the readability increase.

Fig. 1 it could be useful if the reference system of fig 3 is added also in fig. 1. Moreover, the length “b” could be more visible outside of specimen (near “h”) and the roller diameter should be reported.

Fig. 2 it could be useful if the reference system of fig 3 is added. Moreover is not clear if the feed motion is on the specimen or on the tool; the zoom seems to be not coherent, labels pre-processing and processed are correct? Check and/or clarify.

Line 168 “static pressure 1963 N” this is a general problem in the entire paper if you report a pressure the unit should be MPa, if the unit is correct static load should be reported instead that static pressure.

Fig. 3 do you think that the scale of static pressure or load is very useful? This is a general problem of more figures. In my opinion, the scale of static pressure is only an area consuming, but it is your choice. You could use a table with that reports the loads in the first column. However, the surface topographies are not readable because of the scale and label axis are too small. Moreover, all these pictures are really useful to the discussion? Please, consider also the partial overlapping with fig. 9.

Line 176 and paragraph 2.2.3. “the pressure was 300 g” did you mean the mass of the dead weight applied on the indenter? With regard to the paragraph should be specified the geometry and the material of the indenter.

Paragraph 2.2.4 it should be reported the parameters of measurements (current and voltage) anode, filter, voltage, current, peak angle. How was evaluate the depth after layer removal, by measuring only one point or evaluating an average depth like https://doi.org/10.1080/10910340903451472? Please specify.

  • Sections 3: Experimental results

The key point and primary results of this research are the fatigue tests. Surprisingly the direct results of these tests were not reported, the comparison of S-N curves with the measurements is not possible. In my opinion should be reported a graphs and/or tables with the fatigue experiments results.

  • Sections 4: Discussion

In the discussion section, results reported in results section should be discussed with regard to the focus of the paper, fatigue behaviour. All sections and subsections 4.1 (4.1.X) do not reports discussion about the focus of the paper. Probably these sections should be moved in results section.

Lines 379, 397 and Figures 9b, 11, 16: “static pressure”, but it was reported a load. Please check also others similar situations.

Fig. 12 How and where were take TEM specimens? “How” should be reported in Sections 2: materials and methods.

Lines 512, 515: “Berkeley vector” Did you means Burgers vector?

Lines 575-575: “the maximum residual” did you mean the minimum?

Author Response

(The authors gave the same response as above.)

Reviewer 5 Report

I recommend a major revision for the manuscript. There are a lot of mistakes in the manuscript and is very poorly written with a lot of grammatical errors. This manuscript must be thoroughly written that the reader can understand it. I am quoting a few mistakes below.

Line 18: 996.34 mpa must be changed to 996.34 MPa.

Line 19: 'inte-rior' must be changed to 'interior'.

Line 20: 'mate-rial' must be changed to 'material'.         

Line 21: sentence must be changed from ‘change of phase structure’ to ‘changed in the phase structure.’

Line 22: what do you mean by material section?

Line 28: 'Cross sec-tion' must be changed to 'cross-section'.

Paper looks lengthy, few things can go into supporting information.

Author Response

  1. For English language problems, all the grammatical errors mentioned in the article have been corrected and we decided to conduct a detailed vocabulary and grammar check, and submit professional polishing at the end.
  2. The paper has been simplified and highlighted the key content, please review

Round 2

Reviewer 2 Report

I recommend the publication in the present form.

Reviewer 5 Report

The authors have considered the reviewer's comments seriously and have taken care of their grammatical errors. The manuscript reads well now and could be published as-is.